# Coinfection of Cage-Cultured Spotted Sea Bass (*Lateolabrax maculatus*) with *Vibrio harveyi* and *Photobacterium damselae* subsp. *piscicida* Associated with Skin Ulcer

**DOI:** 10.3390/microorganisms12030503

**Published:** 2024-02-29

**Authors:** Dandan Zhou, Binzhe Zhang, Yuchen Dong, Xuepeng Li, Jian Zhang

**Affiliations:** School of Ocean, Yantai University, Yantai 264005, China; m13583064322_1@163.com (D.Z.); lixuepeng20@ytu.edu.cn (X.L.)

**Keywords:** *Lateolabrax maculatus*, *Vibrio harveyi*, *Photobacterium damselae* subsp. *piscicida*, coinfection, extracellular products, pathogenicity

## Abstract

Spotted sea bass (*Lateolabrax maculatus*) is a high-economic-value aquacultural fish widely distributed in the coastal and estuarine areas of East Asia. In August 2020, a sudden outbreak of disease accompanied by significant mortality was documented in *L. maculatus* reared in marine cage cultures located in Nanhuang island, Yantai, China. Two coinfected bacterial strains, namely, NH-LM1 and NH-LM2, were isolated from the diseased *L. maculatus* for the first time. Through phylogenetic tree analysis, biochemical characterization, and genomic investigation, the isolated bacterial strains were identified as *Vibrio harveyi* and *Photobacterium damselae* subsp. *piscicida*, respectively. The genomic analysis revealed that *V. harveyi* possesses two circular chromosomes and six plasmids, while *P. damselae* subsp. *piscicida* possesses two circular chromosomes and two plasmids. Furthermore, pathogenic genes analysis identified 587 and 484 genes in *V. harveyi* and *P. damselae* subsp. *piscicida*, respectively. Additionally, drug-sensitivity testing demonstrated both *V. harveyi* and *P. damselae* subsp. *piscicida* exhibited sensitivity to chloramphenicol, ciprofloxacin, ofloxacin, orfloxacin, minocycline, doxycycline, tetracycline, and ceftriaxone. Moreover, antibiotic resistance genes were detected in the plasmids of both strains. Extracellular product (ECP) analysis demonstrated that both *V. harveyi* and *P. damselae* subsp. *piscicida* can produce hemolysin and amylase, while *V. harveyi* additionally can produce caseinase and esterase. Furthermore, infected fish displayed severe histopathological alterations, including infiltration of lymphocytes, cellular degeneration and necrosis, and loose aggregation of cells. Artificial infection assays determined that the LD_50_ of *P. damselae* subsp. *piscicida* was 3 × 10^5^ CFU/g, while the LD_50_ of *V. harveyi* was too low to be accurately evaluated. Furthermore, the dual infection of *V. harveyi* and *P. damselae* subsp. *piscicida* elicits a more rapid and pronounced mortality rate compared to single challenge, thereby potentially exacerbating the severity of the disease through synergistic effects. Ultimately, our findings offer compelling evidence for the occurrence of coinfections involving *V. harveyi* and *P. damselae* subsp. *piscicida* in *L. maculatus*, thereby contributing to the advancement of diagnostic and preventative measures for the associated disease.

## 1. Introduction

Currently, with the rapid development of intensive farming technology, aquaculture has become the fastest growing food producer in the world. Notably, the spotted sea bass (*Lateolabrax maculatus*) holds substantial commercial worth in both recreational fishery and the mariculture sector [1]. As a euryhaline species, *L. maculatus* is extensively cultivated in East Asia [2]. In recent years, the *L. maculatus* population has encountered escalating disease outbreaks due to the elevated density of cultures and the degradation of water quality, resulting in serious economic losses. Several pathogens, such as *Aeromonas veronii*, hirame rhabdovirus, and *Edwardsiella piscicida*, have been documented as significant contributors to the economic losses experienced by the *L. maculatus* industry [3,4,5].

*Photobacterium damselae* subsp. *piscicida* (formerly known as *Pasteurella piscicida*), a marine bacterium belonging to the family *Vibrionaceae*, stands out due to its extensive host range, high mortality rates, and widespread distribution, resulting in substantial economic losses in marine aquaculture worldwide [6]. *P. damselae* subsp. *piscicida* is the causative agent of pasteurellosis, which was first reported in 1963 in wild populations of *Morone americanus* and *M. saxatilis* [7]. Subsequently, it has been documented that *P. damselae* subsp. *piscicida* possesses the capability to infect a diverse range of marine fish species globally, including *Sparus aurata*, *Dicentrarchus labrax*, *Tursiops truncatus*, *Rachycentron canadum*, and *Trachinotus ovatus* [8,9,10,11]. The clinical manifestations observed in infected fish vary, with certain species displaying hemorrhages in the basal fin and other organs, as well as ulcerative lesions on the skin [12,13], while others exhibit septicemia along with scattered nodules on the spleen and kidney [10,14].

*Vibrio harveyi*, a naturally occurring bacterium in marine environments, has emerged as a formidable pathogen with substantial implications for both wild and cultured marine fish and invertebrates inhabiting warmer waters [15,16]. Diseases caused by *V. harveyi* have led to mass mortality across various fish species, including *Epinephelus fuscoguttatus*, *Epinephelus lanceolatus*, *Paralichthys olivaceus*, and *Sebastes schlegeli*, resulting in significant losses in aquaculture industries [17,18,19]. In fish, outbreaks of *V. harveyi* have been associated with various clinical manifestations, including ocular disease, gastroenteritis, cutaneous lesions, muscle necrosis, and tail rot [20]. Particularly, the presence of skin ulceration is commonly observed in *V. harveyi*-infected fish, leading to the attribution of *V. harveyi* as the primary causative agent of skin ulcer disease [20]. While numerous studies have highlighted the potential pathogenicity of *V. harveyi* in *L. maculatus* [21,22,23], there is currently no documented evidence of natural *V. harveyi* infection in *L. maculatus*.

Bacterial coinfections are a prevalent occurrence in the natural environment, arising from the invasion of multiple distinct pathogens into a single host through simultaneous or secondary infections [24]. Within aquaculture systems, fish commonly encounter a diverse array of opportunistic pathogens, with those possessing broad host ranges being more likely to locate suitable hosts compared to host-specific pathogens [25]. Coinfections can significantly alter the progression and the severity of the diseases, resulting in heightened economic losses. Nevertheless, the impact of coinfections in aquatic animals remains inadequately investigated [24]. There is evidence suggesting that *V. harveyi* is usually found in mixed microbial populations. For instance, *V. harveyi* has been identified alongside *Vibro alginolyticus* and *Vibro parahaemolyticus* in diseased *Pseudosciaena crocea* [26]. Additionally, recent research by Lai et al. has provided evidence of coinfections involving *Cryptocaryon irritans* and *V. harveyi* in orange-spotted grouper [27]. Coinfections involving *P. damselae* and other pathogens are rare, with only one suspected case of coinfection of *P. damselae* subsp. *piscicida* and *Actinomyces*-like organisms observed in *T. truncatus* [9].

In this study, we firstly report a natural bacterial coinfection with *V. harveyi* and *P. damselae* subsp. *piscicida* as a cause of mass mortality in marine cage-cultured *L. maculatus* in Nanhuang island, Yantai, China. Additionally, comprehensive investigations were conducted to evaluate the antibiotic sensitivity, biochemical characterization, genomic functions, extracellular enzymes, histopathology, and pathogenicity of these two bacterial isolates.

## 2. Materials and Methods

### 2.1. Diseased Fish Sampling and Bacterial Isolation

In August 2020, a sudden outbreak of disease accompanied by significant mortality was documented in *L. maculatus* (weighing between 20 and 30 g) that were reared in marine cage cultures located in Nanhuang island, Yantai, China. Diseased fish exhibiting characteristic clinical symptoms were selected, and their spleen and kidney tissues were aseptically extracted for the purpose of disease diagnosis and pathogen identification. The tissues were homogenized using sterilized seawater and subsequently inoculated onto 2216E (Haibo, Qingdao, China) agar plates, which were then incubated at a temperature of 28 °C for a duration of 24 h. Pure cultures of the two strains were obtained after three successive transfers to fresh medium, and stored at −80 °C in marine 2216E medium supplemented with 20% (*v*/*v*) sterile glycerol. The isolates were named NH-LM1 and NH-LM2, respectively.

### 2.2. Molecular Characterization and Phylogenetic Analysis

For 16S rRNA gene and genomic sequencing, genomic DNA of NH-LM1 and NH-LM2 were extracted using a bacterial genomic DNA extraction kit (Tiangen, Beijing, China). The concentrations of purified bacterial DNA were measured with a Nanodrop 2000 spectrophotometer (Thermo Fisher Scientific, Miami, OK, USA) and stored at −20 °C until further analysis. The 16S rRNA gene was amplified by PCR using the universal primers 27F (5′-AGAGTTTGATCCTGGCTCA-3′) and 1492R (5′-GGTTACCTTGTTACGACTT-3′) [28]. DNA sequencing was carried out by Sangon Biotech (Shanghai, China), and the 16S rRNA gene sequence was analyzed using EzBioCloud (http://eztaxon-e.ezbiocloud.net/, accessed on 12 October 2023) [29]. The phylogenetic tree was constructed by neighbor-joining [30] using the Mega software package (v10.0), and the robustness of the phylogenetic trees was estimated by bootstrap analysis based on 1000 replications.

### 2.3. Biochemical and Antimicrobial Resistance Characterization

The enzyme activity and carbon source were determined using the API ZYM and API 20NE systems (bioMérieux, Marcy-l’Étoile, France) according to the instructions. All API tests were performed in triplicate. The susceptibility of NH-LM1 and NH-LM2 to 30 antibiotics listed in Table 1 was tested by the Kirby–Bauer disc diffusion method [31] on 2216E agar plates using commercial antibiotic discs (Hangzhou Microbial Reagent Co., Ltd., HangZhou, China). The antibiotic sensitivity of the strains, as resistant (R), intermediate (I), or sensitive (S), was measured by the diameter of the inhibition zone around each disc according to standards suggested by the company. All tests were performed in triplicate.

### 2.4. Genomic Sequencing and Analysis

The genomes of strains NH-LM1 and NH-LM2 were sequenced using the PacBio Sequel and Illumina NovaSeq PE150 platforms in Novogene Bioinformatics Technology Co., Ltd. (Beijing, China). PacBio reads with low quality (<500 bp) were filtered out, and long reads (>6000 bp) were selected as the seed sequence. The remaining shorter reads were aligned to generate a contiguous sequence without gaps using the SMRT portal assembly software (v5.0.1) [32]. The preliminary assembly result was further corrected using the variant Caller module of the SMRT Link software (v5.0.1) [32], and then refined using Illumina reads aligned by the Burrows–Wheeler Aligner (BWA) (v0.7.12) [33]. The authenticity of the genome was verified using CheckM (Version 1.1.3) [34], and the 16S rRNA gene sequence from the genome was compared with the 16S rRNA gene sequence to ensure its authenticity [35]. The genome component and gene functions were subsequently predicted using the NCBI Prokaryotic Genome Annotation Pipeline (PGAP) [36]. Circular genome maps were generated using Circos to display annotation information [37]. Functional annotations were conducted against 6 databases, i.e., KEGG (Kyoto Encyclopedia of Genes and Genomes), COG (Clusters of Orthologous Groups), NR (Non-Redundant Protein Database databases), Swiss-Prot, GO (Gene Ontology), and TrEMBL. Additionally, the presence of virulence factors was predicted using the Pathogen–Host Interactions database (PHI) and Virulence Factors database (VFDB) [38,39]. Antibiotic resistance genes were analyzed with Antibiotic Resistance Genes Database (ARDB) and the Comprehensive Antibiotic Research Database (CARD) [40,41].

### 2.5. Determination of Extracellular Enzymes and Hemolytic Effect

The extracellular enzymatic activities, including phospholipase, lipase, amylase, hemolysin, and urease, of NH-LM1 and NH-LM2 were assessed following previously established protocols [42,43]. In summary, NH-LM1, NH-LM2, and *Escherichia coli* DH5α were cultured in 2216E media to an OD_600_ of 0.8, then the cells were washed and re-suspended in PBS to 1 × 10^8^ CFU/mL. The 2216E agar plates supplemented with 2% starch, 1% gelatin, 1% starch, and 1% tween-80 were prepared accordingly. Additionally, a sheep blood agar plate (Hope Bio, Qingdao, China) was used for the hemolytic assay. Five microliters of each bacterial suspension were spot-inoculated on the respective plates, and incubated at 28 °C for 24 h. The test was performed in triplicate.

### 2.6. Experimental Infection

#### 2.6.1. Experimental Fish

Clinically healthy *L. maculatus* (average weight 14.7 g) were purchased from a commercial fish farm located in Yantai, Shandong province. The fish were then maintained in aerated seawater at 20 °C. Prior to any experimental procedures, the fish were acclimated in the laboratory for a duration of 2 weeks and verified to be free of pathogenic bacteria. For tissue collection, the fish were euthanized using an excessive dose of tricaine methanesulfonate (Sigma, St. Louis, MO, USA). The live animal research was conducted according to the guidelines of “Regulations for the Administration of Affairs Concerning Experimental Animals” promulgated by Shandong Province. The study, including the experiments involving live animals, was approved by the Ethics Committee of Yantai University.

#### 2.6.2. Single Bacterial Infection and Quantitation of Bacteria in Tissues

For the in vivo infection experiment, strain NH-LM1 or NH-LM2 was cultured in 2216E medium to an OD_600_ of 0.8. The cells were then washed and re-suspended in PBS to obtain various concentrations (ranging from 1 × 10^1^ to 1 × 10^8^ colony-forming units (CFU)/mL). A total of ninety healthy juvenile *L. maculatus* were randomly assigned into nine groups (groups 1 to 9), and intraperitoneally injected with either 100 μL of the bacterial pathogen or PBS. Accumulative mortalities were recorded over a period 14 days, and dead fish were randomly selected for bacteria isolation and 16S rDNA sequencing verification. The lethal dose (LD_50_) values were measured by Karber’s method [44].

For tissue infection analysis, bacterial strains’ NH-LM1 or NH-LM2 were cultured as above and then re-suspended in PBS to achieve concentrations of 10 × LD_50_. A total of fifteen healthy juvenile *L. maculatus* were randomly divided into three groups and i.p. injected with 100 μL of NH-LM1, NH-LM2, or PBS, respectively. At 12 h post-injection (hpi), the fish were euthanized and the bacterial counts in the liver, spleen, kidney, brain, and blood were determined by plate counts.

#### 2.6.3. Coinfection with Strains NH-LM1 and NH-LM1

For coinfection analysis, strains NH-LM1and NH-LM2 were cultured in 2216E until reaching an optical OD_600_ of 0.8, and re-suspended in PBS to 6.0 × 10^7^ CFU/mL and 1.0 × 10^5^ CFU/mL, respectively. A total of eighty healthy juvenile *L. maculatus* were randomly distributed into four groups (groups A to D); group A was challenged with 100 μL of NH-LM1, group B was challenged with 100 μL of NH-LM1, group C was challenged with 100 μL of NH-LM1 and NH-LM2, and group D was injected with 100 μL of PBS. Accumulative mortalities were recorded for 14 d post-infection and the existence of NH-LM1 or NH-LM2 were confirmed by bacterial isolation and 16S rDNA analysis.

### 2.7. Histopathological Observations

*L. maculatus* were challenged with NH-LM1 or NH-LM2 as described above. The dying fish were euthanized using an excessive dose of tricaine methanesulfonate. Subsequently, liver, spleen, kidney, and brain were aseptically collected from the fish, trimmed to the appropriate dimensions, and fixed in 10% (*v*/*v*) neutral buffered formalin for at least 24 h. Additionally, portions of the liver, spleen, kidney, and brain were subjected to plate cultivation and 16S rRNA gene sequencing to confirm that NH-LM1 or NH-LM2 infection was the primary etiological factor of the disease. For histopathological assay analysis, the fixed specimens underwent dehydration through a graded alcohol series and were subsequently embedded in paraffin wax. Subsequently, paraffin sections with a thickness of 5 μm were generated using a Leica™ microtome. These sections were then stained with hematoxylin and eosin (H&E) and examined under a light microscope for observations (Leica DM1000, Hamburg, Germany).

### 2.8. Statistical Analysis

All statistical analyses were performed using the SPSS software (SPSS Inc., Chicago, IL, USA) (v23.0). Survival was estimated by the Kaplan–Meier method and compared by the log-rank test; analysis of variance (ANOVA) was used for another analysis. In all cases, the significance level was defined as *p* < 0.05.

## 3. Results

### 3.1. Clinical Symptoms of Naturally Infected Fish

The naturally infected *L. maculatus* displayed clinical manifestations including anorexia, lethargy, and extensive ulceration on the body surface, with some ulcers penetrating into the muscular tissue. Apart from the ulceration, no additional external symptoms were observed. During necropsy, mild ascites was detected in the abdominal cavity. Certain fish exhibited characteristic clinical lesions, such as hemorrhaging and congestion in the liver, kidney, and intestines, as well as darkening of the spleen, jejunum, and empty stomach (Figure 1).

### 3.2. Isolation and Identification of the Pathogens

After incubating the diseased fish tissue homogenate at a temperature of 28 °C for a duration of 24 h, the presence of two bacterial strains exhibiting distinct morphological and colony size characteristics was observed. These strains were designated as NH-LM1 and NH-LM2. Subsequently, pure cultures of NH-LM1 and NH-LM2 were obtained through three consecutive transfers to fresh 2216E agar plates. To ensure long-term preservation, the strains were stored at a temperature of −80 °C in a marine 2216E broth medium supplemented with 20% (*v*/*v*) glycerol. The identification of the bacterial isolates was confirmed by sequencing the 16S rDNA gene, and the resulting sequences were submitted to GenBank under the accession numbers PP130145 and PP130146. The phylogenetic tree exhibited that strain NH-LM1 was clustered together with known species of *V. harveyi*, and most closely related to strain *V. harveyi* XC141029^T^, meanwhile, strain NH-LM2 was grouped together with known species of *P. damselae* subsp. *piscicida*, and most closely related to strain *P. damselae* subsp. *piscicida* NCIMB 2058^T^ (Figure 2).

### 3.3. Biochemical Characteristics

The biochemical characterizations of *V. harveyi* NH-LM1 and *P. damselae* subsp. *piscicida* NH-LM2 were performed using the API ZYM and API 20NE systems. The results showed that *V. harveyi* NH-LM1 can utilize l-tryptophan, dextrose, l-arginine, urea, aescin iron citrate, gel (bovine source), and 4-nitrobenzene-*β*-d-galactopyranoside; and *P. damselae* subsp. *piscicida* NH-LM2 can utilize dextrose, l-arginine, urea, aescin iron citrate, and 4-nitrobenzene-β-d-galactopyranoside. The API ZYM enzymatic profiles showed that *V. harveyi* NH-LM1 and *P. damselae* subsp. *piscicida* NH-LM2 were both positive for alkaline phosphatase, esterase (C4), esterase lipase (C8), lipase (C14), leucine arylamidase, valine arylamidase, cystine arylamidase, acid phosphatase, naphthol-AS-BI-phosphohydrolase, oxidase, and catalase. In addition, *V. harveyi* NH-LM1 was also positive for trypsin, chymotrypsin, *α*-glucosidase, and N-acetyl-*β*-glucosaminidase (Appendix A).

### 3.4. Antibiotic Sensitivity

The sensitivity of the two strains was assessed using the disc diffusion method on 2216EA plates with antibiotic discs to evaluate their response to 30 antibiotics. The size of the inhibition zone surrounding each disc was used to determine the strains’ antibiotic sensitivity, categorized as resistant, intermediate, or sensitive. The findings revealed that *V. harveyi* NH-LM1 exhibited resistance to 12 antibiotics, including clindamycin, polymyxin B, vancomycin, cefoperazone, cefradine, cefamezin, cephalexin piperacillin, carbenicillin, ampicillin, oxacillin, and penicillin. *P. damselae* NH-LM2 was resistant to 10 antibiotics, i.e., clindamycin, vancomycin, midecamycin, neomycin, gentamicin, erythromycin, carbenicillin, ampicillin, oxacillin, and penicillin. Both strains were resistant to clindamycin, vancomycin, cefamezin, carbenicillin, ampicillin, oxacillin, and penicillin (Table 1).

### 3.5. Genomic Analysis

Genomic analysis indicated that strain *V. harveyi* NH-LM1 contains two circular chromosomes of 2,355,947 bp and 3,694,728 bp (Figure 3), and six circular plasmids of 101,750 bp, 126,749 bp, 57,915 bp, 65,786 bp, 75,229 bp, and 81,481 bp, in which 6136 genes were predicted (Appendix A; Table 2). The predicted numbers of 5S rRNA, 16S rRNA, 23S rRNA, and tRNA sequences were 12, 11, 11, and 129, respectively (Table 2). In addition, 26 genomics islands, two prophages, one CRISPR, and 539 secreted proteins were also detected in the genome of *V. harveyi* (Table 2). Genomic analysis indicated that strain *P. damselae* subsp. *piscicida* NH-LM2 contains two circular chromosomes of 1,302,242 bp and 3,176,321 bp (Figure 4), and three circular plasmids of 114,771 bp, 114,897 bp, and 185,744 bp, in which 4317 genes were predicted (Appendix A; Table 2). The predicted numbers of 5S rRNA, 16S rRNA, 23S rRNA, and tRNA sequences were 21, 19, 19, and 211, respectively (Table 2). In addition, five genomics islands, two prophages, four CRISPR, and 338 secreted proteins were also detected in the genome of *P. damselae* subsp. *piscicida* NH-LM2 (Table 2). The distributions of the COG categories of strains NH-LM1 and NH-LM2 are shown in Figure 3 and Figure 4 and Appendix A.

### 3.6. Pathogenic Genes Analysis

The whole genome of strain NH-LM1 was subjected to comprehensive analysis in order to identify its functional genes associated with pathogenicity. Utilizing the PHI and VFDB databases, the pathogenicity analysis successfully identified 587 and 532 genes related to pathogenicity, respectively. Further classification analysis using the PHI phenotype classification system revealed that the majority of virulence-related genes fell into the categories of reduced virulence (356 genes), loss of virulence (22 genes), and increased virulence (22 genes) (Figure 5). The ARDB database analysis identified 23 genes associated with antibiotic resistance, primarily targeting specific antibiotics of aminoglycoside, macrolide, glycylcycline, penicillin, chloramphenicol, enoxacin, norfloxacin, fluoroquinolone, tetracycline, and trimethoprim. The analysis of the CARD database yielded the identification of 84 genes that are associated with antibiotic resistance, and these findings were corroborated by the alignment of their target antibiotics with the results obtained from the ARDB analysis. Furthermore, the CARD analysis provided insights into the resistance mechanism of NH-LM1, which encompassed antibiotic efflux, reduced permeability to antibiotics, antibiotic inactivation, and alteration of antibiotic targets. Additionally, our investigation also revealed the presence of seven genes encoded by the plasmid that are linked to antibiotic inactivation, with a particular focus on aminoglycosides and cephalosporins.

The pathogenicity genes of NH-LM2 were also analyzed. PHI and VFDB analyses detected 484 and 329 genes related to pathogenicity, respectively. The PHI phenotype classification analysis revealed that the virulence-related genes were mainly classified into the phenotypic mutant types of reduced virulence (287 genes), loss of virulence (21 genes), and increased virulence (27 genes) (Figure 5). Additionally, analysis conducted using the ARDB database identified 21 genes linked to antibiotic resistance, primarily targeting aminoglycoside, macrolide, glycylcycline, beta_lactam, chloramphenicol, fluoroquinolone, streptogramin, tetracycline, trimethoprim, kanamycin, ciprofloxacin, and norfloxacin. The CARD database analysis detected 60 genes associated with antibiotic resistance, and these findings were consistent with the analysis results obtained from the ARDB. Furthermore, the CARD analysis also provided insights into the resistance mechanism of NH-LM2, which includes antibiotic efflux, reduced permeability to antibiotics, antibiotic inactivation, and antibiotic target replacement/alteration. Additionally, we identified six genes encoded by the plasmid that are linked to antibiotic target replacement and antibiotic efflux.

### 3.7. Determination of Extracellular Enzymes and Hemolytic Activities

The hemolytic activity, as well as the presence of lytic enzymes such as caseinase, esterase, amylase, and gelatinase in strains NH-LM1 and NH-LM2 were analyzed using 2216E plates supplemented with the corresponding substrates. The results of the extracellular enzymes and hemolytic activity are presented in Figure 6. NH-LM1 exhibited *β*-hemolysin activity and was capable of producing caseinase, amylase, and esterase, while NH-LM2 demonstrated *α*-hemolysin activity and produced amylase.

### 3.8. Single Experimental Infection with V. harveyi NH-LM1 or P. damselae Subsp. Piscicida NH-LM2

The experimental infection assays showed that mortality was observed in *L. maculatus* infected with either *V. harveyi* NH-LM1 or *P. damselae* subsp. *piscicida* NH-LM2 within 24 h, and reached the peak within 48 h. No diseased or dead fish were observed after the 48 h mark. Additionally, the presence of either *V. harveyi* NH-LM1 or *P. damselae* subsp. piscicida NH-LM2 was confirmed through the re-isolation of the pathogens from the experimentally infected fish, as evidenced by observations of colonial morphology and 16S rDNA sequencing. These findings provide strong evidence that the isolated strains of *V. harveyi* NH-LM1 and *P. damselae* subsp. piscicida NH-LM2 are indeed the causative agents of the *L. maculatus* infection. The results of the pathogenicity study showed that the challenged fish started to die from 12 h, and that 1.0 *×* 10^4^, 1.0 *×* 10^3^, 1.0 *×* 10^2^, and 1.0 *×* 10^1^ CFU/mL of *V. harveyi* NH-LM1 all caused 100% mortality within 48 h; and 9.0 *×* 10^7^, 6.0 *×* 10^7^, 3.0 *×* 10^7^, and 1.0 *×* 10^6^ CFU/mL of *P. damselae* subsp. *piscicida* NH-LM2 caused 100%, 60%, 20%, and 0% mortality within 48 h, respectively (Figure 7). The calculated LD_50_ of *P. damselae* subsp. *piscicida* NH-LM2 to *L. maculatus* was 3 *×* 10^5^ CFU/g, whereas the LD_50_ of *V. harveyi* NH-LM1 could not be calculated. A tissue infection assay showed that both NH-LM1 and NH-LM2 were capable of infecting various organs including the liver, spleen, kidney, brain, and blood. Notably, the spleen and kidney exhibited the highest susceptibility to infection (Figure 8).

### 3.9. Coinfection with V. harveyi NH-LM1 and P. damselae Subsp. Piscicida NH-LM2

In order to validate the occurrence of artificial coinfection, a challenge experiment was conducted using *V. harveyi* NH-LM1 and/or *P. damselae* subsp. *piscicida* NH-LM2 in *L. maculatus*. The cumulative mortality rate of fish challenged with *P. damselae* subsp. *piscicida* NH-LM2 reached 60% within 48 h, while the cumulative mortality rate of fish challenged with *V. harveyi* NH-LM1 reached 100% within 24 h. Remarkably, the coinfected fish experienced a cumulative mortality rate of 100% within 12 h (Figure 9A). The mortality rate associated with coinfection was found to be higher compared to that of single *P. damselae* subsp. *piscicida* NH-LM2 infection, and the time to death was shorter in cases of coinfection compared to single challenges (Figure 9A). Additionally, both *V. harveyi* NH-LM1 and *P. damselae* subsp. *piscicida* NH-LM2 were successfully re-isolated from the head kidney of deceased *L. maculatus*, regardless of whether they were administered as a co-challenge or a single challenge (Figure 9B). In summary, the results indicate that coinfection with *V. harveyi* NH-LM1 and *P. damselae* subsp. *piscicida* NH-LM2 leads to an increased pathogenicity in *L. maculatus*.

### 3.10. Pathological Analysis of Artificially Infected Fish

Compared with the control group, the histopathological examination of *L. maculatus* infected with *P. damselae* subsp. *piscicida* NH-LM2 revealed notable alterations: the liver cells lost their complete polygonal structure, display cell swelling, membrane dissolving, and nuclear fragmentation (indicated by blue arrow), in addition, fibroblast proliferation was also observed (Figure 10B, black arrow); lymphocyte infiltration (black arrow) and cellular vacuolar degeneration (evidenced by nuclear fragmentation and dissolution) (Figure 10E, red arrow) occurred in the spleen; severe necrotic foci were observed in the renal tissue (blue arrow), accompanied by vacuolar degeneration of renal tubular epithelial cells (Figure 10H, black arrow); the tigroid body exhibited dissolution in the central region and marginalization in the periphery, while the glial cells displayed loose aggregation (Figure 10K, black arrow). A histopathological examination of *L. maculatus* infected with *V. harveyi* NH-LM1 showed that the liver cells showed vacuolar degeneration (blue arrow), karyopyknosis, cytoplasmic lysis (Figure 10C, red arrow), and congestion (Figure 10C, black arrow); the presence of blood clots (black arrow) and occurrence of empty areas were observed (Figure 10F, red arrow) in the spleen; the kidney exhibited vacuolar degeneration of renal tubular epithelial cells (black arrow), thickening of the basement membrane of the renal tubules (red arrow), and fibroblast proliferation (Figure 10I, blue arrow); and the brain displayed pathological characteristics including lymphocyte infiltration and loosely aggregated glial cells (Figure 10L, black arrow).

## 4. Discussion

In this study, for the first time, we reported a natural concurrent infection of *V. harveyi* and *P. damselae* subsp. *piscicida* in cage-cultured *L. maculatus*. The diseased *L. maculatus* exhibited pronounced ulceration on the body surface as well as hemorrhage and congestion in the viscus. The outbreak occurred during the premonsoon season, characterized by significant rainfall, fluctuating salinity levels, and elevated water temperature. These environmental factors imposed considerable stress on the cage-cultured fish, rendering them more susceptible to bacterial invasion [45].

*V. harveyi* has been identified as a significant contributing factor to the development of skin ulcers in *Carcharhinus plumbeus*, *Solea senegalensis*, and *E. fuscoguttatus* [46,47,48]. This study also observed ulceration on the body surface and muscle of naturally infected *L. maculatus*. However, no ulceration was observed on the body surface of artificially infected fish, whether they were singly infected or coinfected. Previous studies on *P. damselae* subsp. *piscicida* noted the presence of whitish tubercles in the kidney and spleen of *R. canadum* and *Lates calcarifer* [10,49]. In the present study, the presence of tubercles was not observed in either naturally or artificially infected *L. maculatus*, suggesting that the formation of granulomas may be influenced by the size and species of the infected fish. Pathological examination revealed the occurrence of cytolytic necrosis in the liver, spleen, kidney, and brain, which could potentially be attributed to the release of bacterial toxins such as hemolysin or extracellular enzymes [50]. Similar histological alterations have also been documented in other fish species infected with *P. damselae* subsp. *piscicida*, or *V. harveyi* [50,51].

Previous studies have indicated that the extracellular products of bacteria play a significant role in infection and serve as a direct measure of a bacterial pathogen’s virulence. *V. harveyi*, a known pathogen of aquatic animals, has been found to produce various extracellular products (ECPs), including hemolysins, caseinase, gelatinase, and lipase [20]. Our findings demonstrate that *V. harveyi* NH-LM1 exhibits *α*-hemolysin activity and is capable of producing amylase, caseinase, and lipase. Similarly, *P. damselae* subsp. *piscicida* may also possess extracellular products with potential implications for infection. *P. damselae* subsp. *piscicida* has been recognized as a highly dangerous bacterial pathogen in aquaculture due to its ability to infect a wide range of hosts and cause high mortality rates [52]. One significant virulence mechanism employed by *P. damselae* subsp. *Piscicida* is the production of ECPs, which include cytotoxic, caseinase, lipase, and hemolysins [10,52]. Our results showed that *P. damselae* subsp. *Piscicida* NH-LM2 exhibits *β*-hemolysin activity and is capable of producing amylase. Additionally, a PHI database analysis revealed 356 and 484 genes associated with pathogenicity in NH-LM1 and NH-LM2, respectively. These findings demonstrate the high virulence of strains NH-LM1 and NH-LM2 in cultured *L. maculatus*.

Antibiotics continue to be a cost-effective and efficient means of combating bacterial pathogens, and are widely utilized in numerous countries. However, the widespread use of these medications in clinical settings has led to the selection and dissemination of various antibiotic resistance genes (ARGs) among the microbiota [53]. The antibiotic-sensitivity test revealed that strain *V. harveyi* NH-LM1 was resistant to 12 antibiotics, of which, resistance to cephalosporin (vancomycin, cefoperazone, cefradine) and aminoglycoside (piperacillin) were revealed for the first time [54,55]. Numerous studies have demonstrated that the widespread prevalence of antimicrobial resistance can be partially attributed to the horizontal transfer of antibiotic resistance genes, which is typically facilitated by plasmids [56]. Notably, a genomic analysis of *V. harveyi* NH-LM1 revealed the presence of six plasmids, along with seven genes encoded by plasmid that associated with aminoglycoside and cephalosporin resistance. This finding suggests that antibiotic resistance genes are being horizontally transferred through plasmids. Additionally, the antibiotic-sensitivity test conducted on strain *P. damselae* subsp. *piscicida* NH-LM2 indicated resistance to 10 of the 30 antibiotics, of which, resistance to vancomycin and neomycin, and sensitive to ciprofloxacin, ofloxacin, and doxycycline were revealed for the first time [10,11].

The LD_50_ metric is commonly used to assess the pathogenicity of bacterial pathogens. However, there is currently no available data on the pathogenicity of *V. harveyi* and *P. damselae* subsp. *piscicida* towards *L. maculatus*. *P. damselae* subsp. *piscicida* is known to be a dangerous bacterial pathogen for marine fish, as it exhibits a wide host range and high mortality rate [52]. Wang et al. reported an LD_50_ value of 1.1 × 10^6^ CFU/g for *T. ovatus* [11], while Liu et al. reported an LD_50_ value of 1.03 × 10^4^ CFU/g for *R. canadum* [10]. Turbot were found to be highly susceptible to *P. damselae* subsp. *piscicida*, with a low-dose LD_50_ of ≤1.6 × 10^4^ CFU/fish [49]. In this study, NH-LM2 exhibited significant pathogenicity towards *L. maculatus*, as evidenced by an LD_50_ value of 1.03 × 10^5^ CFU/g. Previous research has also identified *V. harveyi* as a significant pathogen in aquatic animals, particularly fish. Various isolates of *V. harveyi* have been found to display varying degrees of virulence, with LD_50_ values ranging from 2.0 × 10^4^ to 2.53 × 10^7^ CFU/g of fish body weight [17,18,19]. The challenge tests conducted in this study revealed that a concentration of 1 × 10^1^ CFU/fish of NH-LM1 resulted in 100% mortality, indicating that the LD_50_ of NH-LM1 towards *L. maculatus* was too low to be accurately evaluated. Furthermore, the mortality of fish occurred within 12 h of exposure to either of the isolates, suggesting that the infection caused by these pathogens has a short incubation period. Similar phenomena were also reported in *Shewanella algae*-infected tongue soles and *A. veronii*-infected *Carassius auratus*, indicating that the short incubation period contributes to the rapid progression of the disease [57,58]. These findings suggest that both strains NH-LM1 and NH-LM2 exhibit high virulence towards *L. maculatus*, especially the strain NH-LM1, which showed particularly extreme pathogenicity compared to the previously reported *V. harveyi*.

Interactions between invading pathogens have been found to synergistically alter hosts’ susceptibility and duration of infection, leading to an increased pathogenicity of the pathogens [59]. For instance, Chandrarathan et al. reported a synergistic effect in zebrafish coinfected with *A. veronii* and *Aeromnas hydrophila*, resulting in higher mortality rates [60]. Similarly, a study by Xu et al. observed elevated mortality rates in *Micropterus salmoides* coinfected with *A. veronii* and *Nocardia seriolae* [61]. In this study, we observed increased mortality and accelerated mortality in fish coinfected with *V. harveyi* and *P. damselae* subsp. *piscicida* compared to those infected with a single pathogen, thereby indicating the presence of a synergistic effect between *V. harveyi* and *P. damselae* subsp. *piscicida*. These synergistic effects can be attributed to the generation of multiple virulence-related factors and extracellular enzymes, as revealed through genomic and extracellular enzyme analysis. These findings suggest that certain pathogens can mutually benefit from each other, resulting in an intensified amplification of their virulence and ultimately contributing to the occurrence of disease outbreaks in aquatic animals [24].

In this study, the outbreak of the disease was recorded in summer, characterized by significant rainfall, fluctuating salinity levels, and elevated water temperature. Reduced water quality and dramatic shifts in environmental parameters, including turbidity, temperature, oxygen levels, salinity, pH, and organic matter content, are often blamed for infectious disease outbreaks in production aquaculture systems [62,63]. However, in this study, the main environmental factors were not well recorded due to the constraints in sampling tools and conditions. In the future, more investigations and research are imperative to elucidate the intricate interactions among *V. harveyi*, *P. damselae* subsp. *piscicida*, environmental factors, and the fish.

## 5. Conclusions

In this study, for the first time, a dual infection of *L. maculatus* caused by *V. harveyi* and *P. damselae* subsp. *piscicida* was reported and comprehensively investigated. The isolated *V. harveyi* and *P. damselae* subsp. *piscicida* exhibited significant pathogenicity towards *L. maculatus*, and the coinfection of these two bacteria resulted in heightened mortality and rapid mortality in *L. maculatus*. Furthermore, we elucidated the production of ECP, antimicrobial resistance, pathological alterations, and genomic characteristics of these two isolates. These findings contribute novel insights into the pathogenicity of *V. harveyi* and *P. damselae* subsp. *piscicida*, thereby offering valuable knowledge for the diagnosis, prevention, and medical management of fish diseases induced by coinfection with these two bacteria.

## Figures and Tables

**Figure 1 microorganisms-12-00503-f001:**
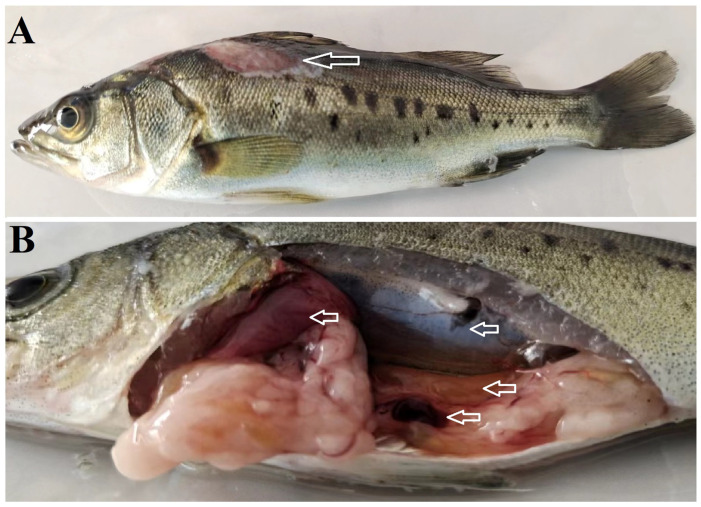
Clinical symptoms of natural coinfection in *Lateolabrax maculatus*. (**A**) Obvious clinical symptoms of natural *Vibrio harveyi* NH-LM1 and *Photobacterium damselae* NH-LM2 coinfection in *L.maculatus* (hollow arrow). (**B**) Visceral lesion features of the diseased fish, arrows indicted the diseased region.

**Figure 2 microorganisms-12-00503-f002:**
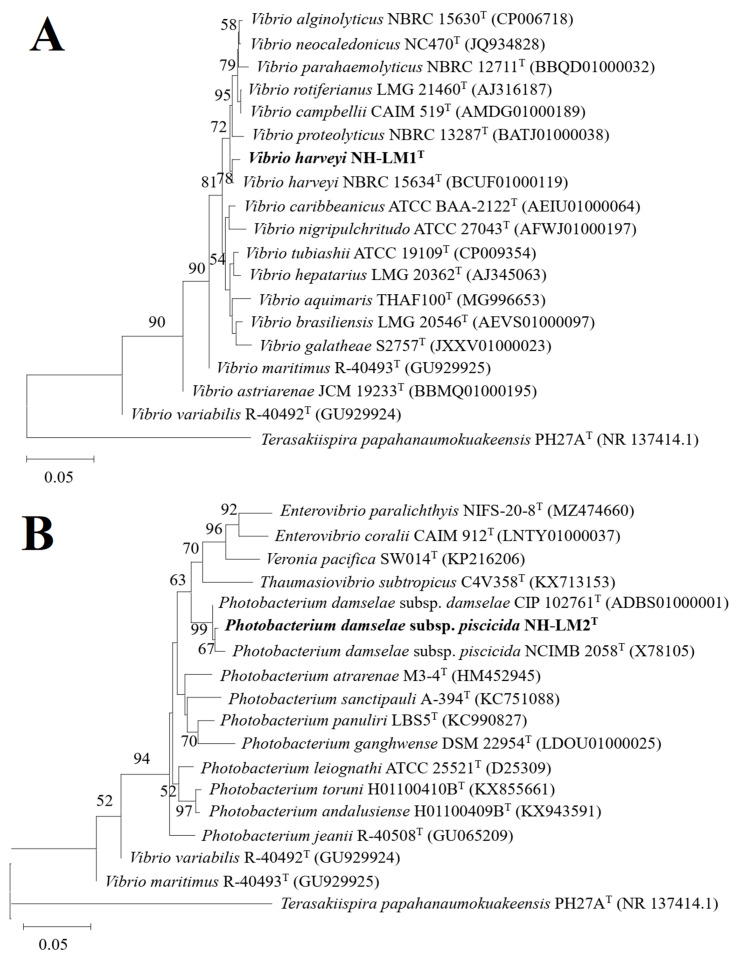
The neighbor-joining phylogenetic tree based on 16S rRNA gene sequences showing the phylogenetic positions of (**A**) *Vibrio harveyi* NH-LM1 and (**B**) *Photobacterium damselae* subsp. *piscicida* NH-LM2. Bootstrap values >50% based on 1000 replications are shown at branching points. *Terasakiispira papahanaumokuakeensis* PH27A^T^ (GenBank accession NR137414.1) was used as an outgroup. Scale bar: 0.05 substitutions per nucleotide position. Bold font indicated the isolated strains of this study.

**Figure 3 microorganisms-12-00503-f003:**
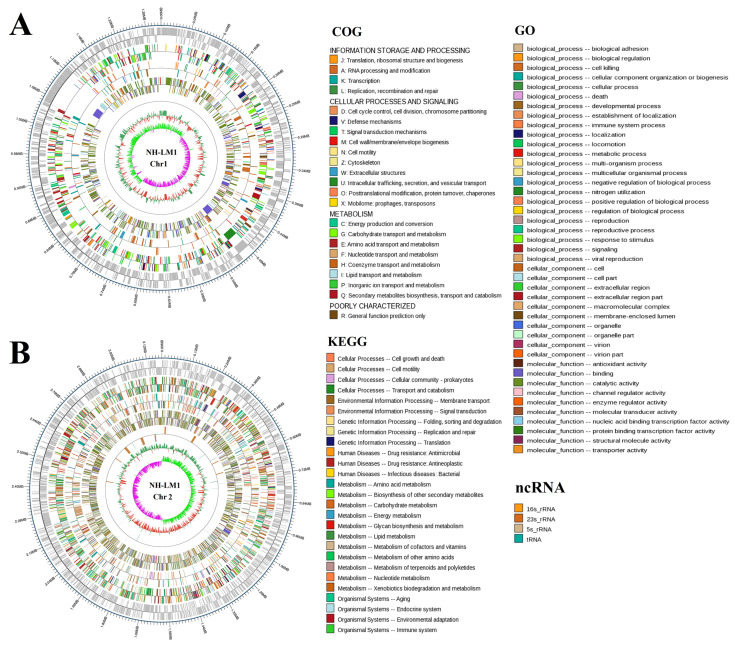
Circular maps of *Vibrio harveyi* NH-LM1 chromosome 1 and chromosome 2. The base pairs are indicated outside the outer circle (circle 1). Circles 1 to 6 represent the coding genes, COG annotation, KEGG annotation, GO annotation, noncoding RNA (ncRNA), and GC content, respectively. (**A**) Chromosome 1, (**B**) Chromosome 2.

**Figure 4 microorganisms-12-00503-f004:**
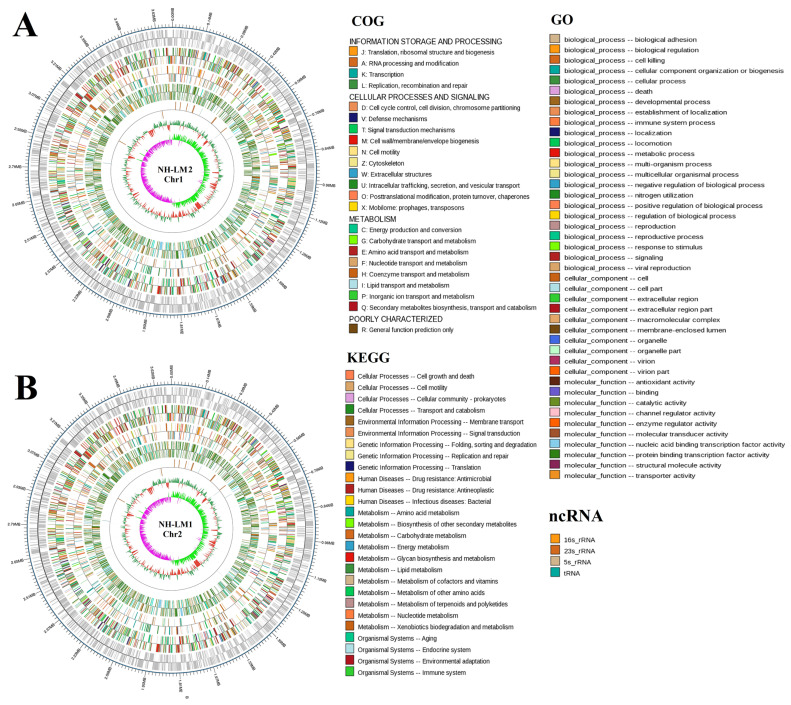
Circular maps of *Photobacterium damselae* subsp. *piscicida* NH-LM2 chromosome 1 and chromosome 2. The base pairs are indicated outside the outer circle (circle 1). Circles 1 to 6 represent the coding genes, COG annotation, KEGG annotation, GO annotation, noncoding RNA (ncRNA), and GC content, respectively. (**A**) Chromosome 1, (**B**) Chromosome 2.

**Figure 5 microorganisms-12-00503-f005:**
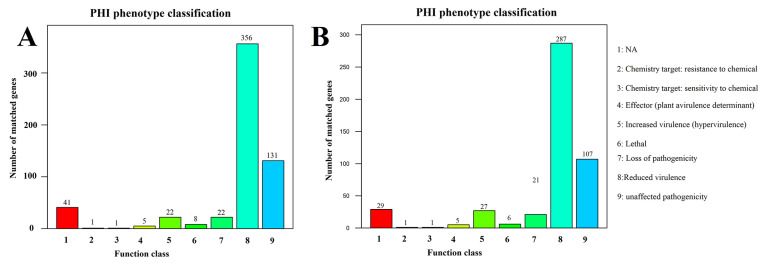
Pathogen host interactions (PHI) annotation of the functional genes of (**A**) *Vibrio harveyi* NH-LM1 and (**B**) *Photobacterium damselae* subsp. *piscicida* NH-LM2. Numbers above each column indicate the gene numbers of different pathogen PHI phenotypic mutant types.

**Figure 6 microorganisms-12-00503-f006:**
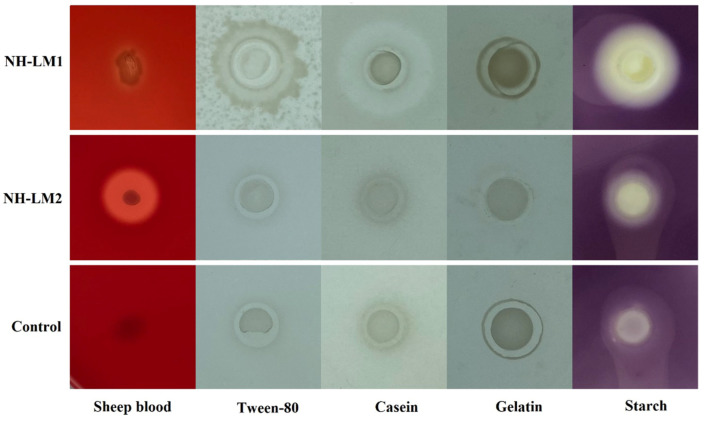
The extracellular enzyme test results of strains NH-LM1 and NH-LM2. The hemolytic activity, as well as the presence of lytic enzymes, including caseinase, esterase, amylase, and gelatinase was analyzed using 2216E plates supplemented with sheep blood, tween-80, casein, gelatin, and starch, respectively. Control: *Escherichia coli* DH5α.

**Figure 7 microorganisms-12-00503-f007:**
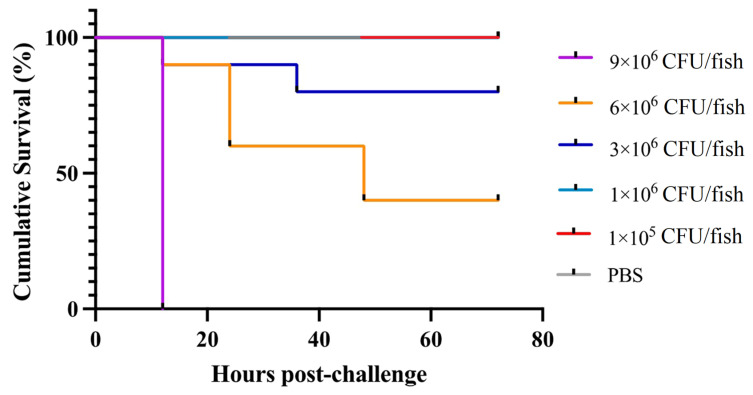
Cumulative survival of *Lateolabrax maculatus* following *Photobacterium damselae* subsp. *piscicida* NH-LM2 infection. *L. maculatus* were challenged with different concentrations of *P. damselae* subsp. *piscicida* NH-LM2 or PBS (control), and the mortalities were observed every 12 h.

**Figure 8 microorganisms-12-00503-f008:**
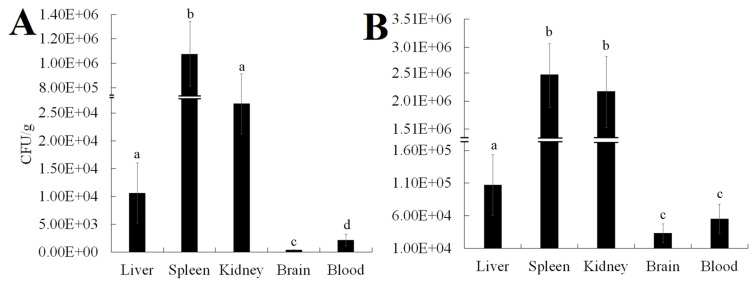
Tissue distributions of *Vibrio harveyi* NH-LM1 or *Photobacterium damselae* subsp. *piscicida* NH-LM2 in infected *Lateolabrax maculatus*. *L. maculatus* artificially challenged with NH-LM1 (**A**) or NH-LM2 (**B**), and the numbers of recovered bacteria in brain, blood, liver, spleen, and kidney were determined. Data are the means of three independent experiments and are presented as means ± SEM. Different letters on the bars denote the statistical significance (*p* < 0.05).

**Figure 9 microorganisms-12-00503-f009:**
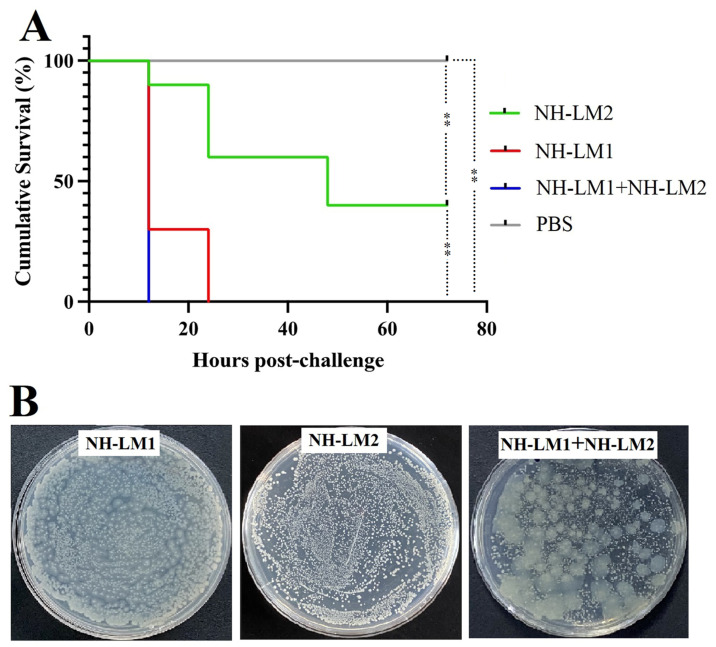
Cumulative survival of *Lateolabrax maculatus* following *Vibrio harveyi* NH-LM1 and/or *Photobacterium damselae* subsp. *piscicida* NH-LM2 infection. *L. maculatus* were challenged with *V. harveyi* NH-LM1, *P. damselae* subsp. *piscicida* NH-LM2, *V. harveyi* NH-LM1 + *P. damselae* subsp. *piscicida* NH-LM2, or PBS (control). The mortalities were observed every 12 h (**A**), and the bacterial recovery was analyzed by plate culture (**B**). Significance was determined with the log-rank test. ** *p* < 0.01.

**Figure 10 microorganisms-12-00503-f010:**
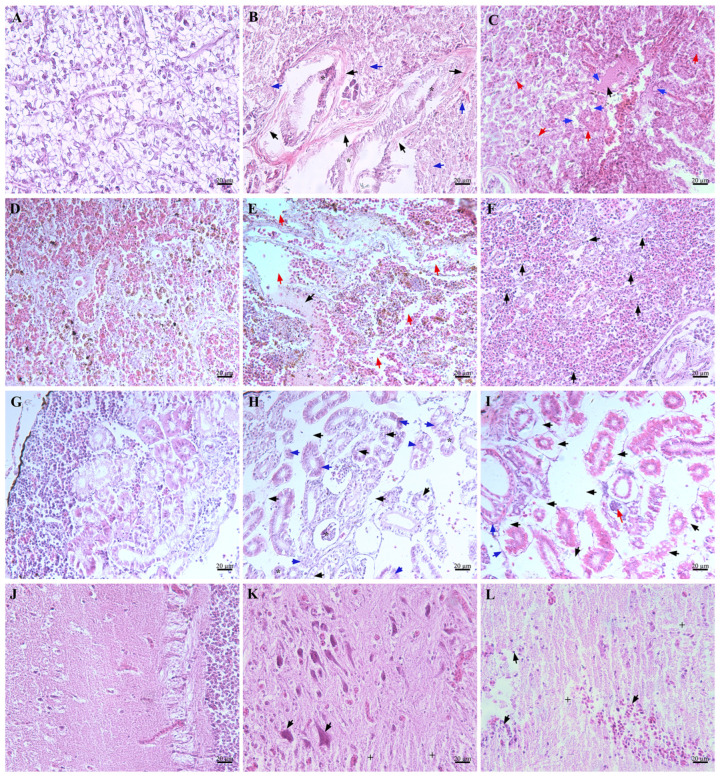
Histological changes in *Lateolabrax maculatus* following *Vibrio harveyi* NH-LM1 or *Photobacterium damselae* subsp. *piscicida* NH-LM2 infection. (**A**,**D**,**G**,**J**) histologic sections of healthy liver, spleen, kidney and brain; (**B**,**E**,**H**,**K**) histologic section of NH-LM2 infected liver, spleen, kidney and brain; (**C**,**F**,**I**,**L**) histologic sections of NH-LM1 infected liver, spleen, kidney and brain. Scale bar = 20 μm. * bacterium, + loosely aggregated glial cells.

**Table 1 microorganisms-12-00503-t001:** Antibiotic susceptibilities of NH-LM1 and NH-LM2.

Antibiotics	Concentration (µg per disc, Unless Otherwise Stated)	NH-LM1	NH-LM2
Clindamycin	2	R	R ^a^
Chloramphenicol	30	S	S
Furazolidone	300	S	I
Polymyxin B	300 IU	R	I
Vancomycin	30	R	R
Ciprofloxacin	5	S	S
Ofloxacin	5	S	S
Norfloxacin	10	S	S
Midecamycin	30	I	R
Erythromycin	15	I	R
Minocycline	30	S	S
Doxycycline	30	S	S
Tetracycline	30	S	S
Neomycin	30	I	R
Kanamycin	30	I	I
Gentamicin	10	S	R
Amikacin	30	S	I
Cefoperazone	75	R	I
Ceftriaxone	30	S	S
Ceftazidime	30	S	I
Cefuroxime	30	I	I
Cefradine	30	R	I
Cefamezin	30	R	R
Cephalexin	30	R	S
Piperacillin	100	R	I
Carbenicillin	100	R	R
Ampicillin	10	R	R
Oxacillin	1	R	R
Penicillin	10 U	R	R

^a^ Abbreviations: R, resistance; I, intermediate; S, sensitive.

**Table 2 microorganisms-12-00503-t002:** Summary of the genome information of strains NH-LM1 and NH-LM2.

Genome Feature	NH-LM1	NH-LM2
Genome size (bp)	6,050,675	4,478,563
Chr1 (bp)	2,355,947	1,302,242
Chr2 (bp)	3,694,728	3,176,321
Plas1	101,750	114,771
Plas2	126,749	114,897
Plas3	57,915	185,744
Plas4	65,786	-
Plas5	75,229	-
Plas6	81,481	-
Encoded genes	6136	4317
Annotated genes	5994	4146
5S rRNA	12	21
16S rRNA	11	19
23S rRNA	11	19
tRNA	129	211
Genomics islands	26	5
Prophages	-	2
CRISPR	1	4
Secreted proteins	539	338

## Data Availability

Data are contained within the article and Appendix A.

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
