# Peer review of "Coinfection of Cage-Cultured Spotted Sea Bass (Lateolabrax maculatus) with Vibrio harveyi and Photobacterium damselae subsp. piscicida Associated with Skin Ulcer"

_microorganisms, 2024, doi:10.3390/microorganisms12030503_

Round 1

Reviewer 1 Report

Comments and Suggestions for Authors

1.       The background present in the abstract is not sufficient. Please update.

2.       The statistical analysis section is missing from material and methods. Please update.

3.       Outgroups are missing in phylogenetic trees. Please include and update the figures.

4.       Figure 6. Legends should be descriptive, including the control details and what is what.

5.       Figure numbers are misleading. Please update. Also, statistical analysis along with p values is a must to present in figures and legends.

6.       What are the future implications of this work? Please discuss in conclusions.

7.       Please include a separate section describing the limitations of the current study.

Author Response

Comment 1: The background present in the abstract is not sufficient. Please update.

Response: The background were added. Page 1, lines 9-11.

Comment 2: The statistical analysis section is missing from material and methods. Please update.

Response: The statistical analysis was added. Page 5, lines 205-208.

Comment 3: Outgroups are missing in phylogenetic trees. Please include and update the figures.

Response: The outgroup strains were added. Figure 2.

Comment 4: Figure 6. Legends should be descriptive, including the control details and what is what.

Response: Detail information was added. Page 12, figure 6.

Comment 5: Figure numbers are misleading. Please update. Also, statistical analysis along with p values is a must to present in figures and legends.

Response: Figure numbers were corrected. Page 15, 16. Statistical analysis was present in figures and legends. Figure 8 and 9.

Comment 6: What are the future implications of this work? Please discuss in conclusions.

Response: The future implications were added. Page 18, lines 528-531.

Comment 7: Please include a separate section describing the limitations of the current study.

Response: The limitations and future research directions were discussed. Page 18.

Reviewer 2 Report

Comments and Suggestions for Authors

This is a very complete study dealing with infections of bacterial pathpgens causing mortality in fish, especially when two bacterial species co-infect the animal. A series of numerous tests regarding the characteristics of the two isolated bacterial strais has been perfored. 

It would be interesting to analyze the environmental factors inducing/facilitating the fish infection, please add a comment on this. These bacterial strains have not been isolated in other farms or China farms?

One mine concern regards the test of antibiotic susceptibility: there is not any indication on the way the sensitivity and resistance have been calculated and valuated, please include this information. Moreover the test has been conducted in the 2216E agar which could not be the most appropiated medium, comment 

line 328: suing??

line 421: E??

Figure 7: I cannot see the lines corresponding to red and PBS

line 444: please control the sentence (of was ECP??)

lines 494-495 and 507-508: causing motility of fish??

Please control along the whole text and include the complete the name of each bacterial and fish species, you cannot used only the first letter (C. irritans, T. truncates.....)

Comments on the Quality of English Language

The quality of English seems to be sufficient

Author Response

Comment 1: It would be interesting to analyze the environmental factors inducing/facilitating the fish infection, please add a comment on this. These bacterial strains have not been isolated in other farms or China farms?

Response:

(1) The comments were added in discussion. Page 18, lines 512-520.

 (2) Until now, Photobacterium damselae subsp. piscicida has not been isolated from Lateolabrax maculatus all over the words. Many studies have highlighted the potential pathogenicity of Vibro harveyi in L. maculatus [21-23], there is currently no documented evidence of natural V. harveyi infection in L. maculatus.

Comment 2: One mine concern regards the test of antibiotic susceptibility: there is not any indication on the way the sensitivity and resistance have been calculated and valuated, please include this information. Moreover the test has been conducted in the 2216E agar which could not be the most appropiated medium.

Response: Relative information was added. Page 3, section 2.3. The marine bacteria strains NH-LM1 and NH-LM2 were isolated and cultured in the 2216E medium, which closely mimics the composition of seawater. Consequently, the antimicrobial resistance analysis was conducted on 2216E agar.

Comment 3: line 328: suing?

Response: Revision has been made. Line 334.

Comment 4: line 421: E??

Response: “E.” represent “Epinephelus”, which has been described in the section of “introduction”, so an abbreviation was used here. Line 61.

Comment 5: Figure 7: I cannot see the lines corresponding to red and PBS

Response: Lines of red and gay were consistent with the line of blue, and we have modified the figure to show these three lines. Figure 7.

Comment 6: line 444: please control the sentence (of was ECP??)

Response: Revision has been made. Line 453

Comment 7: lines 494-495 and 507-508: causing motility of fish??

Response: Revision has been made. Lines 489, 504.

Comment 8: Please control along the whole text and include the complete the name of each bacterial and fish species, you cannot used only the first letter (C. irritans, T. truncates.....)

Response: The full name of T. truncates was first present in line 53. The full name of some fish and bacterial species were added. If the name was not appeared as the first time, an abbreviation of the genus name was used. Lines 79, 81, 501,502.

Round 2

Reviewer 1 Report

Comments and Suggestions for Authors

The authors successfully responded to the reviewers comments and updated the manuscript as well.